# A universal metabolite repair enzyme removes a strong inhibitor of the TCA cycle

Anthony J. Zmuda[1,3], Xiaojun Kang[1,3], Katie B. Wissbroecker[1], Katrina Freund Saxhaug[2], Kyle C. Costa[1], Adrian D. Hegeman [1,2] & Thomas D. Niehaus [1] ✉

A prevalent side-reaction of succinate dehydrogenase oxidizes malate to enol-oxaloacetate (OAA), a metabolically inactive form of OAA that is a strong inhibitor of succinate dehydrogenase. We purified from cow heart mitochondria an enzyme (OAT1) with OAA tautomerase (OAT) activity that converts enol-OAA to the physiological keto-OAA form, and determined that it belongs to the highly conserved and previously uncharacterized Fumarylacetoacetate_hydrolase_domain-containing protein family. From all three domains of life, heterologously expressed proteins were shown to have strong OAT activity, and ablating the *OAT1* homolog caused significant growth defects. In *Escherichia coli*, expression of succinate dehydrogenase was necessary for OAT1-associated growth defects to occur, and ablating *OAT1* caused a significant increase in acetate and other metabolites associated with anaerobic respiration. OAT1 increased the succinate dehydrogenase reaction rate by 35% in in vitro assays with physiological concentrations of both succinate and malate. Our results suggest that OAT1 is a universal metabolite repair enzyme that is required to maximize aerobic respiration efficiency by preventing succinate dehydrogenase inhibition.

Many enzymes catalyze promiscuous side reactions that form noncanonical compounds, some being toxic if allowed to accumulate[1,2]. Metabolite damage control systems exist to limit the accumulation of noncanonical compounds; this often involves a metabolite repair enzyme that reconverts a damaged metabolite to a physiological form[3]. Most enzymes of the TCA cycle are known to catalyze promiscuous reactions and two iconic cases of metabolite damage and repair have been described here[4]. Both involve reduction of 2-oxoglutarate by side reactions of either malate dehydrogenase (MDH) or variants of isocitrate dehydrogenase to form L-2-hydroxyglutarate (L2HG) or D-2-hydroxyglutarate (D2HG), respectively[5–7]. L2HG and D2HG are noncanonical metabolites in most species that inhibit various enzymes[6,8]. Distinct dehydrogenases (DH) that are each highly conserved across the three domains of life reconvert these damaged metabolites back to 2-oxoglutarate[9–11]. Disruption of either L2HGDH or D2HGDH can cause distinct and severe heritable metabolic diseases in humans[12–15].

Another major promiscuous activity associated with the TCA cycle is catalyzed by succinate dehydrogenase (SDH), which plays key roles in the TCA cycle and electron transport chain by coupling the oxidation of succinate to fumarate with the reduction of a quinone to a quinol (Fig. 1). SDH also readily oxidizes malate to the noncanonical enol form of oxaloacetate (OAA)[16,17]. Enol-OAA is not a substrate for any known transporters or OAA-metabolizing enzymes, all of which use the physiological keto-OAA form exclusively[18]. Additionally, OAA inhibits SDH with the enol form being much more potent than the ketone[16,19–21]. Enol-OAA will spontaneously tautomerize to keto-OAA, but the rate of interconversion is slow[18,22,23]. A few enzymes with enol-keto-OAA tautomerase (OAT) activity have been identified. Bovine aconitase and *Escherichia coli* fumarase, both TCA cycle enzymes, have demonstrated OAT side-activities[24,25]. However, OAT activity of bovine aconitase is maximum at pH 9.0 and drops sharply below that, being reduced by >95% at pH 7.0[26]. A bovine enzyme (denoted OAT1) whose

[1]Department of Plant and Microbial Biology, University of Minnesota, Twin Cities, Saint Paul, MN 55108, USA. [2]Department of Horticultural Science, University of Minnesota, Twin Cities, Saint Paul, MN 55108, USA. [3]These authors contributed equally: Anthony J. Zmuda, Xiaojun Kang. ✉e-mail: tniehaus@umn.edu

OAT activity at neutral pH is ~170-fold more catalytically efficient than that of bovine aconitase was reported decades ago, but its sequence was unknown[26]. We predicted that OAT1 plays an important role in OAA tautomerization in vivo and sought to determine the molecular identity of this potential metabolite repair enzyme.

Here, we identify the highly conserved gene encoding OAT1 and show that this enzyme is required to prevent SDH inhibition under physiological conditions. We also show that deleting OAT1 causes growth phenotypes and TCA cycle attenuation in vivo, suggesting that OAT1 plays a critical role in aerobic respiration.

## Results

### Molecular identification of OAT1

OAT1 was partially purified from bovine heart mitochondria using a combination of centrifugal and ammonium sulfate fractionations followed by mixed-mode column chromatography. A sample was obtained that was judged to be 51% pure OAT1 based on OAT-specific activity (Fig. S1). The addition of 50 µM oxalate, a specific inhibitor of OAT1[26], reduced OAT activity in this sample >90%. SDS-PAGE analysis showed that one apparent ~32-kD protein predominated the sample (Fig. S1). This band was excised from the gel and subject to trypsin digest and mass spectroscopic analysis. The data indicated that the band consisted of two proteins, <u>F</u>umarylacetoacetate <u>h</u>ydrolase <u>d</u>omain-containing protein 2 A (FAHD2A) and FAHD2B, that are encoded by separate genes and differ only by a single amino acid (Val or Leu at position 243) at the protein level (Fig. S2).

The FAHD annotation is based on sequence homology to the C-terminal conserved domain of fumarylacetoacetate hydrolase (FAH), which catalyzes the final step in Phe and Tyr degradation (Fig. 2A)[27]. FAHD and the C-terminal domain of FAH are structurally conserved, consisting of three layers of beta-sheets arranged in a unique 'FAH' fold[28,29]. FAH and FAHD are distinct members the 'FAA hydrolase' protein family (pfam01557) that also includes some less-common enzymes that are involved in specialized metabolism, such as *E. coli* 2-hydroxypentadienoate hydratase (MhpD) and *Saccharolobus solfataricus* 2-keto-3-deoxy-D-arabinonate dehydratase (KdaD; Fig. 2A). MhpD and KdaD have C-terminal FAH-like domains and, like FAH, catalyze reactions with substrates and products that can exist as enol- or keto-tautomeric forms (Fig. 2A), and further, enol-keto tautomerization plays a critical role in their catalytic mechanisms[28,30,31]. FAH proteins are conserved in animals but occur sparsely in fungi, bacteria, and archaea, while FAHD proteins are essentially completely conserved across the three domains of life. The human genome encodes

three FAHD proteins: FAHD2A and FAHD2B that have 98% conserved sequence identity (fig. S3) and are mitochondrial localized, and FAHD1 that lacks ~90 N-terminal amino acids relative to FAHD2 and has been identified in the cytosol and mitochondria[32] (Fig. 2A). The ~90 N-terminal amino acid extension appears to play a role in protein: protein interactions that influence localization in *B. subtilis*[33]. The genomes of multicellular eukaryotes often encode more than one FAHD protein, sometimes with and without N-terminal extensions (Fig. 2A).

Despite their widespread occurrence, the function of FAHD proteins was unclear. The *Bacillus subtilis* homolog YisK plays a role in remodeling the bacterial peptidoglycan envelope[34] and knocking out the *Saccharomyces cerevisiae* homolog FMP41 decreases stress resistance[35]. Studies have linked FAHD function to aerobic respiration in *Caenorhabditis elegans* and human cells, for which FAHD mutants have inhibited mitochondrial electron transport chain function[36–38]. Genetic co-essentiality mapping strongly associates FAHD1 with the TCA cycle in mammalian cells[39]. The only FAHD enzymes with reported molecular functions that may be physiologically relevant are human FAHD1 and *B. subtilis* YisK, which were shown to have minor OAA decarboxylase activity[33,40]; human FAHD1 has an even more modest acylpyruvase activity[41]. *FAHD* genes cluster on the chromosome with genes predicted to encode L2HGDH and/or D2HGDH in several taxonomically diverse bacteria (Fig. 2B), which could indicate that all three have TCA cycle-related metabolite damage repair roles. However, no conserved function for FAHD enzymes has been described.

### FAHD enzymes have OAT activity

Since FAHD enzymes are widely conserved and our proteomics data indicate that bovine FAHD2A and/or FAHD2B is the orphan enzyme OAT1, several diverse FAHD proteins were selected to test for OAT activity (Fig. S3). The coding sequences of bovine FAHD2A and homologs from human (FAHD1, FAHD2A, and FAHD2B), *Arabidopsis thaliana* (FAHD1 and FAHD2), *Saccharomyces cerevisiae* (FMP41), *E. coli* (YcgM), *Bacillus subtilis* (YisK), and *Methanococcus maripaludis* (FAHD) were cloned into the pET28b vector to facilitate expression of mature enzymes with N-terminal polyhistidine tags. We also cloned a gene encoding human FAHD1 mutated at two residues (denoted M2-FAHD1) predicted to facilitate substrate binding[40,41] (Fig. S4). Recombinant enzymes were produced in *E. coli* and purified to near homogeneity (Fig. S5).

The chemical properties of OAA make it well-suited for enol-keto tautomerase assays. The enol form of OAA is a fully conjugated molecule and strongly absorbs UV light ($\varepsilon_{260} = 11\,mM\,cm^{-1}$), unlike the unconjugated keto form[26]. OAA exists predominantly as a ketone at equilibrium in aqueous solution (87% at pH 9.0 and 93.5% at pH 2.0)[26]. Conversely, the enol tautomer predominates at equilibrium when OAA is dissolved in an aprotic solvent such as diethyl ether or acetone (~65% in dry solvent)[26]. When OAA in diethyl ether was added to a pH 9.0 buffered solution, the slow spontaneous tautomerization of enol-OAA to keto-OAA as the mixture approached equilibrium could be directly observed by monitoring absorbance at 260 nm (Fig. 3A). Addition of 1 µg of *E. coli* YcgM dramatically increased the rate of tautomerization and quickly brought the mixture to equilibrium (Fig. 3A). All FAHD enzymes that were purified had similar effects except for M2-FAHD1, which had <0.1% of the OAT activity as non-mutated human FAHD1 (Fig. S4). These results indicate that all FAHD enzymes have high OAT activity in the physiologically relevant enol→keto direction. Although some crystal structures of FAHD enzymes show a bound $Mg^{2+}$ or $Ca^{2+}$ that is likely involved in substrate binding[28,42], addition of divalent cations or EDTA had no effect on OAT activity in our recombinant enzyme preparations (Fig. S6) or for native bovine OAT1[26], suggesting that either the metal is tightly bound or that its presence is variable across isoforms. The specific activity of recombinant bovine FAHD2A

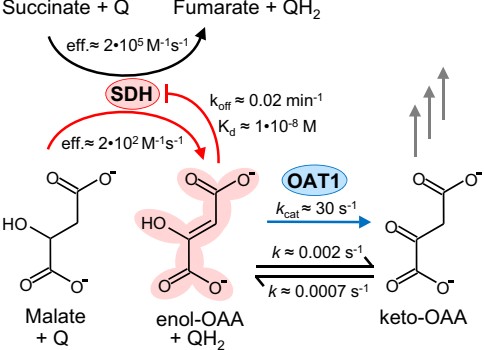

**Fig. 1 | Formation and removal of an inhibitory side-product of the TCA cycle.** Canonical and promiscuous activities of SDH are indicated with black and red arrows, respectively. Enol-OAA, a strong inhibitor of SDH, is converted to keto-OAA spontaneously (⇌) or by OAT1 (blue arrow). All values shown were reported at pH 7.0 and 25 °C, and those associated with SDH or OAT1 were reported for *Bos taurus* enzymes[16,21,26]. Q quinone, QH₂ quinol, eff. catalytic efficiency ($k_{cat}/K_M$), $k_{off}$ dissociation rate constant, $K_d$ equilibrium dissociation constant, $k_{cat}$ turnover number, $k$ rate constant.

was almost identical to that reported for native bovine OAT1 under similar assay conditions[26], further corroborating that bovine FAHD2A/B is the orphan enzyme OAT1.

To directly assess OAT activity in the keto→enol direction, aqueous OAA solution at pH 2.0 was added to a pH 9.0 buffered solution and the slow spontaneous tautomerization of keto-OAA to enol-OAA was monitored (Fig. 3B). Addition of 1 μg of *E. coli* YcgM increased the rate of tautomerization, albeit at a lesser-rate than in the enol→keto direction (Fig. 3B). The same trend was observed for other non-mutated FAHD enzymes.

**A**

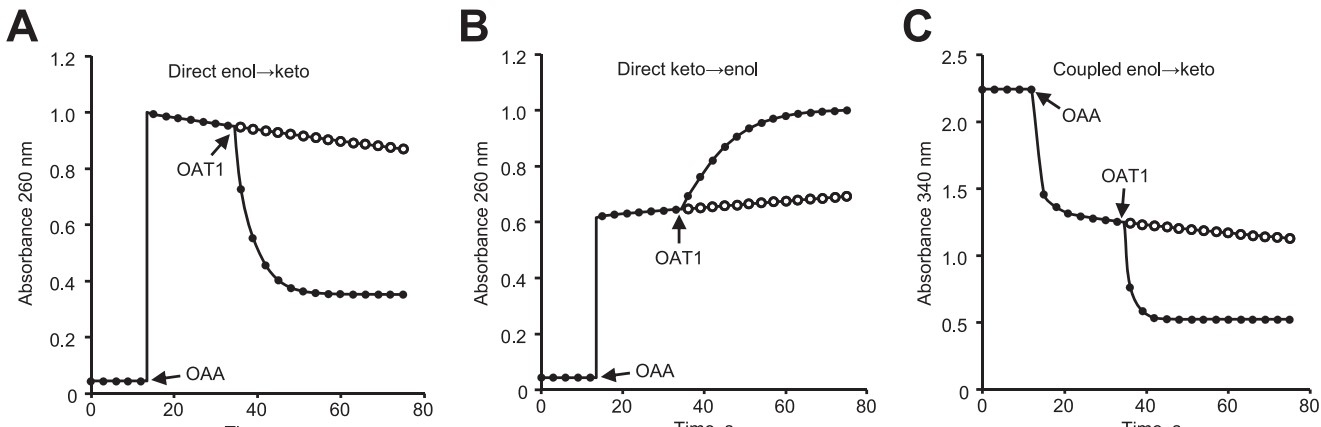

**Fig. 2 | Phylogenetic relationships, domain architectures, and reactions catalyzed by FAH hydrolase family enzymes. A** Phylogram of representative FAH family enzymes. Nodes with bootstrap values ≥0.5 are shown. Typical architectures of each subfamily of enzymes is shown with conserved domains colored as indicated. Reactions catalyzed by each subfamily of enzymes is shown with enol-keto tautomers highlighted. *BS Bacillus subtilis, Mm Methanococcus maripaludis, Dr Danio rerio, Hs Homo sapiens, Sc Saccharomyces cerevisiae, Ec Escherichia coli, Dm Drosophila melanogaster, Ce Caenorhabditis elegans, At Arabidopsis thaliana, Ss Saccharolobus solfataricus.* **B** Predicted FAHD genes (colored blue) cluster with predicted L2HGDH (colored orange) and/or D2HGDH (colored purple) genes in various configurations in several diverse prokaryotic genomes. Apparently unrelated genes are colored white.

**Fig. 3 | Representative OAT assays. (A)** OAA in diethyl ether or **(B)** aqueous OAA (pH-2) was added to pH 9 buffered solution. **C** OAA in acetone was added to pH 7.5 buffered solution containing MDH and NADH. One μg *E. coli* YcgM (OAT1) was added as indicated. Open circles show progression of the assays without added enzyme.

The activity measured in either direct OAT assay will always be negatively influenced by the reverse reaction due to the fact that both substrates, enol and keto-OAA, are always present. A MDH coupled assay avoids this problem by rapidly converting keto-OAA to malate so that activity against enol-OAA can be measured in the absence of the ketone. When OAA in acetone was added to a pH 7.5 buffered solution containing MDH and NADH, a rapid drop in absorbance at 340 nm was followed by a slow decrease in absorbance (Fig. 3C). This trend corresponds to the rapid MDH-mediated oxidation of keto-OAA present in the sample, followed by the oxidation of keto-OAA as it slowly forms via spontaneous tautomerization of enol-OAA. The addition of 1 μg of *E. coli* YcgM caused the absorbance at 340 nm to drop rapidly (Fig. 3C) and addition of other non-mutated FAHD enzymes gave similar results. The total amount of NADH oxidized equaled the amount of OAA added to the assay. As expected, the measured OAT activity in the enol→keto direction was slightly greater in the coupled assay than the direct assay (Fig. 3A, C). Apart from confirming that FAHD enzymes have high OAT activity in the enol→keto direction, this coupled assay nicely shows that MDH is only active against keto-OAA, and further, that enol-OAA is relatively long-lived in solution – even in the presence of OAA-utilizing enzymes – unless an enzyme with OAT activity is present. Kinetic parameters of the various FAHD enzymes were determined in the enol→keto direction using the coupled assay (Table 1). FAHD enzymes showed a range of kinetic parameters, which is not uncommon for enzymes conserved across kingdoms[43], yet even the least catalytically efficient enzyme (Arabidopsis FAHD2) was in the range of physiological relevance[44].

Although human FAHD1 and *B. subtilis* YisK reportedly have OAA decarboxylase activity[33,40], we did not observe significant decarboxylase activity in direct OAT assays; after reactions containing FAHD reached equilibrium, the absorbance at 260 was stable for several minutes (Fig. 3), indicating that the total concentration of OAA was unchanged. Still, we felt it prudent to assess the OAA decarboxylase activity of all recombinant FAHD enzymes. Using a direct assay in which the substrate and product, OAA and pyruvate, could be accurately quantified by HPLC, slight OAA decarboxylase activity was measured for each FAHD enzyme (Table 1). The average ratio of OAT to OAA decarboxylase specific activities for FAHD enzymes is ~$2 \times 10^4$, indicating that OAA decarboxylase is a minor side-activity of FAHD compared to OAT activity.

## OAT1 prevents SDH inhibition by enol-OAA

Enol-OAA is a potent inhibitor of SDH[16,19–21], which suggests that a major physiological role of OAT1 is removing enol-OAA to prevent SDH inhibition. To test whether OAT1 can impact SDH activity, SDH was partially purified from bovine heart submitochondrial particles for analysis[45,46]. SDH can be assayed colorimetrically using a reducing dye such as 2,6-dichlorophenolindophenol[45]. Because this colorimetric assay is based on reducing equivalents generated by SDH, oxidation of both succinate and malate can be effectively measured. Additionally, glutamate-oxaloacetate transaminase (GOT) and glutamate were included in the assay to prevent keto-OAA accumulation, as would normally occur in vivo.

Our SDH preparation had a specific activity of 0.18 μmol min$^{-1}$ mg$^{-1}$ when assayed with 2 mM succinate, which is nearly the same activity reported previously[45]. The addition of 10 μg of human FAHD1 (or equivalent units of other FAHD enzymes) had no effect on enzyme activity, nor did addition of the catalytically inactive M2-FAHD1 (Fig. 4A), confirming that OAT activity has no effect on the canonical reaction of SDH. When SDH was assayed with 2 mM malate, activity was low and approached zero within a few minutes (Fig. 4B). The Addition of 10 μg of FAHD1 (or equivalent units of other FAHD enzymes) increased SDH activity and allowed the reaction to reach a steady-state, but not when M2-FAHD1 was included (Fig. 4B). These results confirm that malate oxidation by SDH forms enol-OAA, which GOT is unable to remove until it tautomerizes to the ketone, and that FAHD-dependent removal of enol-OAA prevents SDH inhibition. Since similar concentrations of both succinate and malate are typically available to SDH in vivo[47,48], SDH was assayed with 2 mM of each potential substrate. Under these conditions, SDH activity was 35.5% less than the rate determined with succinate alone (Fig. 4C). Addition of 10 μg of FAHD1 (or equivalent units of other FAHD enzymes) caused the rate to increase to its maximum of 0.18 μmol min$^{-1}$ mg$^{-1}$, but not when M2-FAHD1 was included (Fig. 4C). These results indicate that enol-OAA formation by SDH is significant under physiological conditions and that OAT activity practically eliminates SDH inhibition that would otherwise be unavoidable.

## Phenotypic consequences of OAT1 disruption

Our in vitro data indicating that FAHD enables SDH to operate at its maximum rate under normal cellular conditions is corroborated by the results of in vivo studies in *C. elegans* and human cell lines. The genome of *C. elegans* encodes a single FAHD protein, FAHD1 (Fig. 2A). A deletion mutant of *FAHD1* has significantly reduced mitochondrial membrane potential and reduced O$_2$ consumption compared to wild type[36]. Impaired mitochondrial function results in reduced body and brood size and severe locomotion deficit[36]. Similarly, knocking down FAHD1 in human umbilical vein endothelial cells by shRNA-mediated RNA interference results in significantly reduced mitochondrial membrane potential, basal oxygen consumption, and spare respiratory capacity, which causes reduced cell proliferation and premature senescence[37]. It was recently shown that lentiviral knock-down of FAHD1 in the breast cancer cell lines MCF-7 and BT-20 results in lower SDH activity[38], strongly indicating that enol-OAA inhibition occurs in vivo when OAT

**Table 1 | Kinetic properties of FAHD enzymes**

| Enzyme | OAA tautomerase (enol→keto) | | | | OAA decarboxylase[a] |
|---|---|---|---|---|---|
| | $v_{max}$ (μmol min$^{-1}$ mg$^{-1}$) | $K_M$ (μM) | $k_{cat}$ (s$^{-1}$) | $k_{cat}/K_M$ (M$^{-1}$ s$^{-1}$) | Specific activity (μmol min$^{-1}$ mg$^{-1}$) |
| *Bt* FAHD2A | 53.5 ± 4.8 | 20.2 ± 1.4 | 32.7 ± 3.0 | $1.63 \times 10^6$ | 0.008 ± 0.001 |
| *Hs* FAHD1 | 69.0 ± 10.0 | 15.8 ± 5.9 | 31.1 ± 4.5 | $1.99 \times 10^6$ | 0.011 ± 0.001 |
| *Hs* FAHD2A | 22.2 ± 4.5 | 22.2 ± 3.6 | 13.6 ± 2.8 | $6.28 \times 10^5$ | 0.004 ± 0.001 |
| *Hs* FAHD2B | 112 ± 5.3 | 296 ± 12 | 68.7 ± 3.3 | $2.32 \times 10^5$ | 0.022 ± 0.003 |
| *At* FAHD1 | 8.4 ± 1.1 | 48.8 ± 1.8 | 3.7 ± 0.5 | $7.48 \times 10^4$ | 0.008 ± 0.001 |
| *At* FAHD2 | 1.7 ± 0.2 | 66.0 ± 4.9 | 0.7 ± 0.1 | $1.10 \times 10^4$ | 0.003 ± 0.001 |
| *Sc* FMP41 | 717 ± 80 | 219 ± 23 | 369 ± 41 | $1.75 \times 10^6$ | 0.006 ± 0.002 |
| *Ec* YcgM | 372 ± 65 | 26.9 ± 0.4 | 161 ± 28 | $5.95 \times 10^6$ | 0.011 ± 0.001 |
| *Bs* YisK | 402 ± 13 | 43.8 ± 8.1 | 237 ± 8 | $5.81 \times 10^6$ | 0.064 ± 0.002 |
| *Mm* FAHD | 201 ± 18 | 24.8 ± 2.2 | 83.2 ± 7.4 | $3.38 \times 10^6$ | 0.299 ± 0.008 |

[a]OAA decarboxylase activity was determined at 1.0 mM OAA.

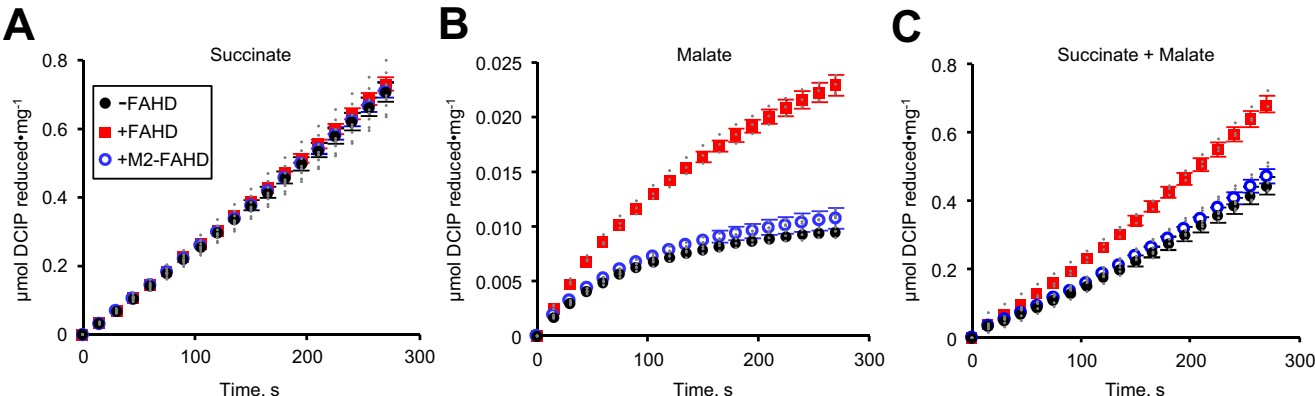

**Fig. 4 | FAHD increases SDH activity at physiological substrate concentrations.** Bovine SDH activity at 2 mM succinate (**A**), 2 mM malate (**B**), or 2 mM each succinate and malate (**C**). Assays contained 1 mM MOPS-NaOH, pH 7.1, 5 μg (1 U) GOT, 1 mM glutamate, 0.1 mM DCIP, 0.1 mM decylubiquinone, detergent-solubilized SDH (11–33 μg total protein), and substrate(s) without (black circles) or with addition of 10 μg *H. sapiens* FAHD1 (red squares) or 10 μg M2-FAHD1 (open blue circles). DCIP reduction was monitored at 600 nm. Data represents the mean and SEM, $n = 3$ independent experiments; individual data points are represented with gray circles. In assays with malate and succinate, the difference in mean steady-state rates determined without and with FAHD1 was 35.5% and this difference is significant (2-sided $t$ test; $P = 0.0021$, $t = 7.065$, df = 4, 95% CI = −0.001403 to −0.0006114).

activity is reduced. In addition to demonstrating OAT activity being required for efficient aerobic respiration, these results indicate that FAHD enzymes are the major source of OAT activity in mitochondria.

The phenotypic consequences of disrupting FAHD activity in animals is severe[36,37], as is the case for the TCA cycle-associated metabolite repair enzymes L2HGDH and D2HGDH[12–15]. However, disruption of either L2HGDH or D2HGDH has little effect on microbial growth[49,50]. We analyzed *FAHD* knockouts of *Saccharomyces cerevisiae*, *Escherichia coli*, and *Methanococcus maripaludis*. In all three organisms, deletion mutations in their single *FAHD* gene caused significant growth defects relative to wild type, and wild-type growth was restored to mutants complemented with their respective plasmid-expressed *FAHD*. Growth of an *E. coli FAHD* mutant was impaired on plates containing LB medium or standard M9 minimal medium (supplemented with glucose) (Fig. 5A). The growth defect was more severe when the minimal medium was supplemented with the respiratory substrates glycerol or OAA (Fig. 5A). The mutant also showed reduced growth rate in liquid standard M9 medium (Fig. 5B) and reduced colony size on solid LB medium (Fig. 5C). The yeast mutant had impaired growth on liquid SC medium supplemented with OAA (Fig. 5D), and had smaller colonies on solid YPG medium (Fig. 5E). In *M. maripaludis*, which utilizes the reductive TCA cycle[51], the mutant also had reduced growth rate (Fig. 5F). Thus, disruption of *FAHD* causes significant growth impairments in organisms spanning the three domains of life, even when grown under ideal culture conditions.

Additional *E. coli* mutants were used to probe the mechanisms causing growth defects in *ycgM* deficient cells. We obtained a *ΔsdhA E. coli* strain that lacks the catalytic subunit of SDH, and created a *ΔsdhAΔycgM* double mutant strain. *ΔsdhA* colonies on LB medium were slightly smaller than those of wild type, and wild-type growth was restored to *ΔsdhA* cells complemented with plasmid-expressed sdhA (Fig. 5C). The colony size of *ΔsdhAΔycgM* double mutant cells was slightly less than *ΔsdhA* cells, but interestingly, the phenotype was not as severe as observed in *ΔycgM* cells (Fig. 5C). Complementing the double mutant with *ycgM* caused a slight increase in colony size to the level of *ΔsdhA* cells, however, complementing with *sdhA* significantly decreased the colony size of the double mutant to the level of *ΔycgM* cells (Fig. 5C). These results demonstrate that the colony size growth phenotype observed in *ΔycgM* cells is dependent on having a functional SDH, confirming a role for OAT1 in removing a toxic byproduct produced by SDH. Further, these results suggest that promiscuous oxidation of malate by SDH is the most physiologically important source of enol-OAA in vivo, at least in *E. coli*.

## Metabolic impacts of OAT1 disruption

A metabolomics analysis was performed on *E. coli* cells to explore metabolic signatures associated with *ycgM* deletion during both mid-log (Fig. 5G and Supplementary Data 1) and early stationary (Fig. S7) growth phases. Parallel samples were grown in M9 minimal medium with either glucose of natural isotopic abundance or fully labeled [$^{13}$C6] glucose, which provided additional information for confirmation of LC-MS feature elemental composition and metabolic origin[52]. By mixing the labeled and unlabeled samples, this approach controls for matrix differences providing added confidence in relative abundance measurements. Dramatic changes in the relative abundance of TCA cycle intermediates were not apparent during mid-log growth, although modest statistically significant abundance differences were measured for fumarate, succinate, and (iso)citrate (Fig. 5G). Most notably in other parts of metabolism, all of the detected intermediates of the Entner–Doudoroff (ED) and pentose phosphate pathways were significantly elevated in the *ycgM* mutant (Fig. 5G). This was also true for the early steps in the Embden–Meyerhoff–Parnas (EMP) pathway, and for pyruvate, acetyl-CoA, and acetate, which was highly elevated in *ycgM* mutant cells (Fig. 5G). This accumulation is likely due to decreased catabolic carbon flux through the TCA cycle. Utilization of the ED pathway is thought to balance energy production with protein costs under conditions (e.g. TCA cycle inactivity) where glucose utilization via EMP is inefficient due to the requirement for high concentrations of its enzymes to overcome a buildup of glycolytic products[53]. The observed accumulation of acetate is also consistent with carbon being diverted from entering the TCA cycle via overflow metabolism[54]. Together, these metabolomics results are consistent with a significantly attenuated TCA cycle in the *ycgM* mutant. Metabolic flux profiling of *E. coli* SDH/MDH deficient and wild-type cells showed an increase in ED glucose flux from trace levels in wild type to 30% in mutant cells[55], consistent with our observations. The same study also showed that the percentage of OAA produced through the TCA cycle dropped from 44% in wild type to effectively 0% in the SDH/MDH mutant, such that all OAA was derived from phosphoenolpyruvate (PEP) through PEPCK, as is typical for the TCA cycle functioning anaplerotically. In the *ycgM* mutant, the TCA cycle also appears to be functioning in biosynthesis rather than in a catabolic capacity. The significant decrease in PEP in the mutant (Fig. 5G) suggests that PEPCK is compensating for reduced SDH activity by supplying carbon to the back-end of the TCA cycle. Anaplerosis could also account for why succinate doesn't accumulate in the *ycgM* mutant, but a metabolic flux analysis is needed to confirm this interpretation. The metabolic

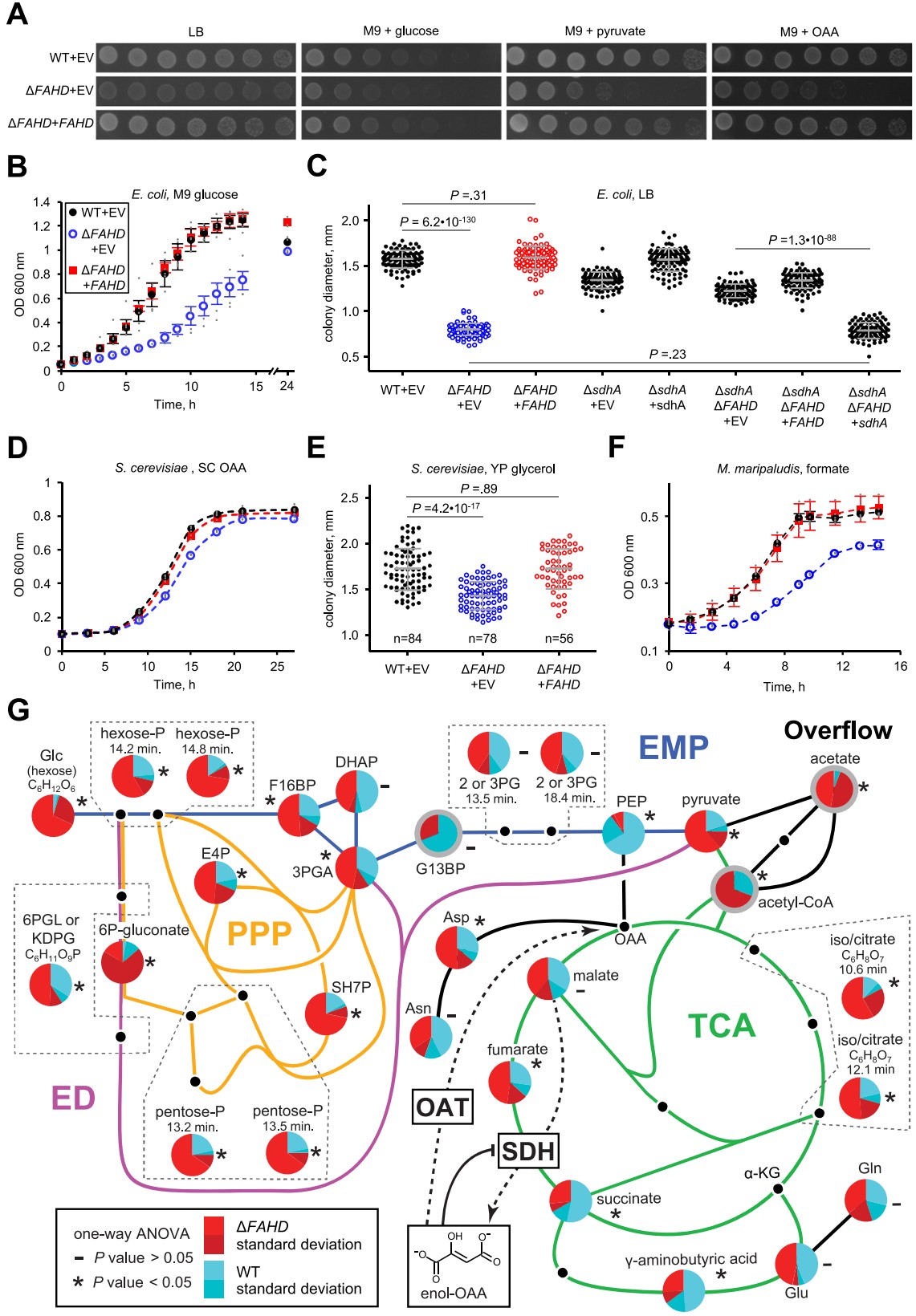

perturbations were generally not as pronounced in early stationary phase when energy demand diminished (Fig. S7). Many intermediates of the early EMP, ED, and pentose phosphate pathways, as well as acetate, were less abundant in early stationary compared to mid-log phase and the differences in relative abundance between wild type and mutant were reduced (Fig. S7).

## Discussion

It is well established that promiscuous activities of TCA cycle enzymes can form enzyme effectors that cause severe metabolic disorders[13,15]. Malate oxidation by SDH to form enol-OAA is likely to be the most promiscuous reaction of the TCA cycle yet discovered. The ratio of the canonical reaction's catalytic efficiency to the promiscuous reaction's

**Fig. 5 | Phenotypic consequences of FAHD ablation. A** Serially diluted ΔycgM (ΔFAHD) or wild type *E. coli* containing either *ycgM*-pUC19 (*FAHD*) or empty vector (EV) were spotted on the indicated solid medium plates and pictures were taken after an incubation period. These experiments were repeated three times with similar results. **B**, **D**, **F** Growth of the wild-type strain with empty vector or ΔFAHD strain with empty vector or complemented with *FAHD*-vector for the indicated organism grown in the indicated liquid medium. Data represents mean and SEM, *n* = 3 biologically independent cell cultures. **C**, **E** Colony size of the indicated *E. coli* or *S. cerevisiae* strains on the indicated solid medium. Data represents mean and SD, n = 95 (*E. coli*) or the indicated (*S. cerevisiae*) biologically independent cell colonies. 2-sided *t* tests were used to determine whether differences in means are significant, with *P* values shown for key comparisons (other statistical information is listed in the source data). **G** The relative abundance of respiratory intermediates in ΔycgM (red slices) or wild-type (blue slices) *E. coli* determined by isotope ratio HILIC-MS analysis. Pie charts outlined in gray show relative abundance determined by comparing peak heights. Metabolite ID ambiguities are enclosed by dotted black lined polygons with chromatographic retention times listed where multiple peaks were detected for different isomers. Data represents mean and SD, *n* = 4 biologically independent cell cultures given two treatments. Statistical significance was determined with one-way ANOVA (*P* values are listed in Supplementary Data file 1). EPM Embden–Meyerhoff–Parnas pathway, blue, PPP pentose phosphate pathway, orange, ED Entner–Doudoroff pathway, magenta, TCA tricarboxylic acid cycle, green.

catalytic efficiency for SDH ($\sim 10^3$)[16] is several orders of magnitude more efficient than the promiscuous reaction of MDH ($10^7$–$10^8$ for the human enzyme)[5] that forms L2HG. Accordingly, our results show that the promiscuous oxidation of malate to enol-OAA by SDH in vivo is inevitable (Fig. 4). We further show that OAT activity of FAHD enzymes can practically eliminate enol-OAA inhibition of SDH that would otherwise occur indefinitely (Figs. 3 and 4). Disrupting FAHD in all domains of life causes significant growth defects (Fig. 5), which have been linked to aerobic respiration in animals[36–38], and specifically to SDH in *E. coli* (Fig. 5C) and human[38]. The metabolic profile of *E. coli* indicates a severely attenuated TCA cycle consistent with SDH inhibition (Fig. 5G). Together, these results confirm the physiological occurrence of this metabolite damage repair pathway and demonstrate its critical importance to the TCA cycle and aerobic respiration.

Although protecting SDH from inhibition is apparently the major function of FAHD enzymes, there is evidence for additional roles. OAA features in many metabolic pathways, is present in multiple subcellular compartments in eukaryotes, and its buildup can be deleterious[56,57]. FAHD enzymes may be involved in maintaining low OAA concentrations throughout the cell by allowing the rapid removal of both tautomers. The fact that the FAH-fold is structurally conserved among FAHD enzymes and other members of the FAA hydrolase protein family that act on diverse enol-keto tautomerizable metabolites indicates that FAHD enzymes may have additional catalytic functions. This is supported by comparative genomics; prokaryotic *FAHD* genes often cluster on the chromosome with genes encoding enzymes involved with enol-keto tautomerizable metabolites (Fig. S8), in addition to clustering with the TCA cycle metabolite damage repair genes *L2HGDH* and *D2HGDH* (Fig. 2B). We also demonstrated acetoacetate enol-keto tautomerase activity for FAHD enzymes (Fig. S9), confirming that they can act on at least one metabolite other than OAA. The conservation of the FAH-fold also suggests that its ancestral catalytic function is enol-keto tautomerization.

OAT1 joins L2HGDH and D2HGDH as ubiquitous metabolite repair enzymes functioning to remove harmful side-products of the TCA cycle, underscoring the prominence of metabolite damage in the TCA cycle and the importance of its repair[58,59]. It is likely that metabolite damage and its repair or preemption is widespread throughout metabolism and characterizing these instances will be necessary for completing metabolic networks[60]. Because OAT1 significantly affects the efficiency of the TCA cycle, its role in energy metabolism and organismal physiology should be investigated further. Many metabolic pathways feature OAA or other enol-keto tautomerizable intermediates, including some synthetic pathways such as carbon-fixing C4-glyoxylate cycles[61], and OAT enzymes could be a valuable tool for improving the efficiency of these pathways.

## Methods
### Chemicals and reagents
Oxaloacetic acid and anhydrous diethyl ether were from ACROS Organics. Acetone was from Fisher Chemical. Acetone was dried by mixing 1 g of anhydrous calcium sulfate with 100 mL acetone in a sealed container and incubating overnight; acetone was dried daily as needed. Malic dehydrogenase (MDH) from porcine heart and glutamic-oxaloacetic transaminase (GOT) from porcine heart were from Sigma Aldrich. Hydroxyapatite was from Bio-Rad. All other chemicals were from Sigma Aldrich.

### Purification and identification of bovine OAT1
All procedures were performed at 4 °C or on ice. Bovine heart mitochondria were prepared according to a standard protocol[45]. We began with ~1 kg of trimmed meat from a freshly harvested steer heart. The final yield was ~1 g of total protein as determined by Bradford dye-binding assays. Isolated mitochondria were suspended in 50 mL of 10 mM Tris-HCl, pH 7.5, 250 mM sucrose and stored at −80 °C.

OAT1 was partially purified based on previously published methods[26]. 50 mL of frozen mitochondria suspension was thawed overnight and diluted with 50 mL of 50 mM potassium phosphate (KPi), pH 8.1, 2 mM ethylenediaminetetraacetic acid (EDTA). 25 mL fractions were sonicated using a Braun-Sonic 2000 set to 100% power for five, 15 s pulses, cooling on ice for 2 min between pulses. The mixture was centrifuged for 15 min at 10,000×*g*. The supernatants were collected, diluted with an additional 50 mL of 50 mM KPi, pH 8.1, 2 mM EDTA, and centrifuged for 60 min at 100,000×*g*. The clear supernatant was collected and ammonium sulfate was slowly added to 35% (w/v) while maintaining the pH at 7.8-8.0 with concentrated ammonium hydroxide. The mixture was stirred for 30 min and centrifuged for 15 min at 10,000×*g*. The clear supernatant was collected and the salt concentration was brought to 70% (w/v) ammonium sulfate while maintaining pH at 7.8−8.0. After 30 min of stirring, the sediment was collected by centrifugation and the pellet resuspended in 20 mL of 80% (w/v) ammonium sulfate, 5 mM KPi, pH 7.8, 0.2 mM EDTA, 5 mM DTT. The suspension was dialyzed with cellulose membrane tubing (~12,000 MWCO) against 1000 mL of dialysis buffer (0.2 mM KPi, pH 7.8, 0.2 mM EDTA, 1 mM DTT) for 16 h, exchanging with fresh buffer every 4 h. The cloudy solution was centrifuged for 15 min at 10,000×*g* and the clear supernatant was collected.

Clarified supernatant was applied to a 20 mL hydroxyapatite column equilibrated with dialysis buffer. The column was first washed with 200 mL of dialysis buffer. While collecting 5 mL fractions, proteins were eluted with 75 mL of 0.2 mM KPi, pH 7.8, followed by 30 mL of 1.0 mM KPi, 30 mL of 5 mM KPi, and 30 mL of 10 mM KPi. The six fractions eluted at 10 mM KPi, were concentrated using 10,000 MWCO centrifugal filter units and used in Bradford dye-binding assays, OAT activity assays (described below), and SDS-PAGE analysis. The predominant band judged to be bovine OAT1 was excised from the SDS-PAGE gel and submitted for trypsin digest and MS analysis by the University of Minnesota Center for Mass Spectrometry & Proteomics. The data was processed with Sequest set up to search uniprot_bos_taurus9913_UP0000091320200622_unipr_cRAP.fasta.

### Comparative genomics analysis
Sequences were taken from the UniProtKB database or the SEED database. Analysis of protein families and conserved domains was

performed with the NCBI Conserved Domain Database (CDD) and EMBL-EBI Protein Families Database (Pfam). Comparative analysis of 984 representative genomes was performed with SEED and its tools; the full results of the analysis are available in the SEED subsystem named "FAHD". To construct phylogenetic trees, sequences were aligned with MUSCLE, curated with Gblocks, phylogeny determined with PhyML, and tree rendering by TreeDyn.

## Cloning FAHD genes

Sequences encoding *Bos taurus* FAHD2A (UniProt ID: F1MLX0), *Homo sapiens* FAHD1 (UniProt ID: Q6P587, splice variant 1), *Homo sapiens* FAHD2A (UniProt ID: Q96GK7), *Homo sapiens* FAHD2B (UniProt ID: Q6P2I3), *Arabidopsis thaliana* FAHD1 (UniProt ID: Q93ZE5), *Arabidopsis thaliana* FAHD2 (UniProt ID: Q9LUR3), *Saccharomyces cerevisiae* FMP41 (UniProt ID: N1NWG6), *Methanococcus maripaludis* FAHD (UniProt ID: A0A7J9S1W3), *Escherichia coli* YcgM (UniProt ID: P76004), and *Bacillus subtilis* YisK (UniProt ID: O06724) were cloned to facilitate expression of mature proteins with N-terminal hexahistidine tags. Coding sequences were PCR amplified from either genomic DNA (*S. cerevisiae, M. maripaludis, E. coli,* and *B. subtilis*), cDNA synthesized from total RNA obtained from calf heart or leaf (*B. taurus* and *A. thaliana*) or synthetic DNA (*H. sapiens*). For synthetic *H. sapiens* sequences, FAHD1 was the native coding sequence (isoform 1), FAHD2A was the native coding sequence except for two silent mutations included to remove a long stretch of cytosine nucleotides (Fig. S10), FAHD2B was codon optimized to enhance bacterial expression (Fig. S10), and M2-FAHD1 contained three nucleic acid substitutions causing D102A R106A mutations (Fig. S10). Primers are listed in Table S1. Amplicons were treated with the restriction enzymes NdeI and XhoI, except for that of *B. subtilis* (NheI and XhoI) and *S. cerevisiae* (NheI and EcoRI), and were ligated into the matching sites of pET28b. For complementation studies, the *E. coli ycgM* and *sdhA* coding sequences were cloned into pUC19 via HindIII and XbaI, the *S. cerevisiae* Fmp41 coding sequence and native promotor (including the 606 nucleotides upstream of the coding sequence in the genome) was cloned into pRS425 via HindIII and BamHI, and the *M. maripaludis* FAHD coding sequence was cloned into pLW40neo via NsiI and AscI. All constructs were verified by Sanger sequencing.

## Protein expression and purification

*E. coli* strain BL21-(DE3)-RIPL (or Rosetta-gami B(DE3) for expressing human FAHD2B) harboring an expression construct was grown in 200 mL LB medium containing 50 μg/mL kanamycin at 37 °C with shaking until $OD_{600}$ reached 0.8. Cultures were cooled to 22 °C, and isopropyl β-D-thiogalactoside (IPTG) and ethanol were added to 0.5 mM and 4% (v/v), respectively. Cultures were shaken overnight at 22 °C. Cells were harvested by centrifugation (4000 ×*g*, 10 min) and resuspended in 7 mL of cold lysis buffer (50 mM Tris-HCl, pH 8.0, 300 mM NaCl, 10 mM imidazole). Subsequent steps were performed at 4 °C or on ice. The suspension was sonicated with a Braun-Sonic 2000 set to 50% power for six, 15 s pulses, waiting 60 s between pulses to cool the samples. Lysate was centrifuged at 20,000×*g* for 10 min and the supernatant was added to a column containing 0.2 mL of HisPur Ni-NTA resin (Thermo Fisher Scientific) and washed with 20 mL of wash buffer (50 mM Tris-HCl, pH 8.0, 300 mM NaCl, 20 mM imidazole). Recombinant proteins were eluted with 0.5 mL of elution buffer (50 mM Tris-HCl, pH 8.0, 300 mM NaCl, 200 mM imidazole). Centrifugal filter units (10,000 MWCO) were used to concentrate proteins and exchange buffer with 100 mM KCl, 50 mM Tris-HCl, pH 8.0. Glycerol was added to 10% (v/v) and 10–20 μL aliquots were snap-frozen in liquid nitrogen and stored at −80 °C. Freshly thawed aliquots were used in subsequent experiments.

## OAT enzyme assays

**Direct enol→keto.** OAA solutions were made fresh daily by dissolving OAA in dry diethyl ether or dry acetone. Assays (0.2 mL, pH 9.0)

contained 2 mM Tris-HCl, in a 0.1 mL cuvette at 22 °C. While measuring absorbance at 260 nm on an Agilent Cary 3500 spectrophotometer using Cary UV Workstation software version 1.1.298, 2 μL of 20 mM oxaloacetic acid in dry diethyl ether was added, rapidly mixed, and 100 μL was transferred to another 0.1 mL cuvette to monitor the reaction in parallel. One μg of *E. coli* OAT1 (5 μL of a 0.2 μg/ μL solution in 2 mM Tris-HCl, pH 9.3) or mock was added and rapidly mixed. The difference in rates measured in the two parallel reactions was determined to be the rate of enzyme-catalyzed tautomerization.

**Direct keto→enol.** Assays (0.2 mL, pH 9.0) contained 2 mM Tris-HCl in a 0.1 mL cuvette at 22 °C. While measuring absorbance at 260 nm, 8 μL of 20 mM oxaloacetic acid (pH-3) was added, rapidly mixed, and 100 μL was transferred to another 0.1 mL cuvette to monitor the reaction in parallel. One μg of *E. coli* OAT1 (5 μL of a 0.2 μg/ μL solution in 2 mM Tris-HCl, pH 9.3) or mock was added and rapidly mixed. The difference in rates measured in the two parallel reactions was determined to be the rate of enzyme-catalyzed tautomerization.

**Coupled enol→keto.** Assays (0.2 mL, pH 7.5) contained 10 mM potassium phosphate, 10 μg (12 U) malate dehydrogenase, 0.3 mM NADH in a 0.1 mL cuvette at 22 °C. While measuring absorbance at 340 nm, 2 μL oxaloacetic acid (2-20 mM) in dry acetone was added, rapidly mixed, and 100 uL was transferred to another 0.1 mL cuvette to monitor the reaction in parallel. 0.01–1.0 μg of enzyme or mock was added to the first half reaction and rapidly mixed. The difference in rates measured in the two parallel reactions was determined to be the rate of enzyme-catalyzed tautomerization. Kinetic parameters were calculated by fitting data to the Michaelis–Menten equation using GraphPad Prism software version 5.01.

## SDH enzyme assays

Submitochondrial particles (SMPs) were created from isolated bovine heart mitochondria according to published procedures[45,46]. SMP's were diluted to 10 mg mL$^{-1}$ total protein in 10 mM Tris-SO$_4$, pH 7.4, 0.25 M sucrose, and 1 mL aliquots were snap frozen in liquid N$_2$ and stored at −80 °C. To solubilize and enrich succinate dehydrogenase (SDH), an aliquot of SMPs was thawed on ice and the suspension was centrifuged 82,000 *g* for 30 min. The pellet was resuspended in 1 mL of 10 mM Tris-SO4, pH 7.5 using a Dounce homogenizer. *n*-Dodecyl β-D-maltoside was added to a final concentration of 0.5% (w/v) and incubated for 30 min at 4 °C with shaking. The mixture was centrifuged at 48,000 *g* for 30 min. The supernatant consisted of solubilized SDH and was held on ice and used in assays within 2 h.

Assays (0.1 mL) contained 1 mM MOPS-NaOH, pH 7.1, 5 μg (1 U) glutamate-oxaloacetate transaminase, 1 mM L-glutamate, 0.1 mM DCIP, 0.1 mM decylubiquinone, either 2 mM succinate, 2 mM malate, or 2 mM each succinate and malate, and without or with 10 μg human FAHD1, M2-FAHD1, or equivalent units of other recombinant FAHD enzymes, in a 0.1 mL cuvette at 22 °C. While measuring absorbance at 600 nm, 5 μ–15 μL (5 μ–33 μg total protein) of purified SDH from bovine heart SMPs was added and rapidly mixed.

## OAA decarboxylase assays

Assays (0.1 mL, pH 7.5) contained 10 mM potassium phosphate, 1 mM freshly prepared OAA, 0.5 mM MgCl$_2$, and were started by adding 5 μL of enzyme storage buffer containing either no enzyme (mock) or 5 μg of enzyme. Freshly thawed enzyme aliquots were used for all assays. Assays were incubated at 22 °C for 30 min before being stopped by adding 5 μL of 1 M HCl and placing on ice. We verified that the levels of OAA and pyruvate in stopped reactions remained unchanged for several hours at 4 °C. 50 μL of stopped reaction was analyzed with an Agilent 1100 series HPLC with a G1315B DAD detector using a BioRad Aminex HPX-87H 300 mm × 7.8 mm column with 5 mM H$_2$SO$_4$ as the mobile phase (1 mL min$^{-1}$ flow rate). Compounds were detected by

absorbance at 200 nm. The amount of compound in each peak was determined by integrating peak areas using OpenLab ChemStation software version 2.19.20, and comparing values to those obtained from standard curves prepared for OAA and pyruvate. OAA decarboxylase enzyme activity was determined by subtracting the amount of pyruvate formed in mock reactions (i.e., spontaneous decarboxylation) from that formed in reactions with added enzyme.

## Strains and culture conditions

*S. cerevisiae* knockout strain YNL168C (BY4741 FMP41p::*kan*MX4) and the parental strain BY4741 (*MAT* a; *his3Δ1;leu2Δ0;met15Δ0;ura3Δ0*) were provided by Horizon Discovery. The knockout strain was PCR-verified before use (Fig. S11). Strains were transformed with pRS425-FMP41+promoter or empty vector using the Yeastmaker™ Yeast Transformation System 2 (Takara) and transformants were selected on minimal synthetic dropout (SD) medium lacking Leu. For growth curves, freshly transformed strains were grown overnight in SD medium lacking Leu, washed once, and used to inoculate to an initial $OD_{600} = 0.1$, 5 mL of SD medium lacking Leu in which glucose was replaced with 10 mM OAA and 2% (v/v) ethanol. cultures were grown at 28 °C with shaking and $OD_{600}$ was recorded every 3 h for 27 h. To determine colony size on plates, freshly transformed strains were grown overnight in SD medium lacking Leu and aliquots of serially diluted culture expected to contain ~100 cells (based on $3 \times 10^7$ cell mL$^{-1}$ $OD_{600}^{-1}$) were spread on plates containing YPD medium in which glucose was replaced with 2% (w/v) glycerol. Plates were incubated at 22 °C for five days and high resolution images were acquired with a Biorad Gel Doc XR Imaging System. ImageJ software version 1.53t was used to measure the size of colonies for each strain using the 'analyze particles' function ($n = 83$, WT + EV; 78, ΔFAHD + EV; 55, ΔFAHD + FAHD).

*E. coli* Keio collection *ΔycgM* strain (designation JW1169), *ΔsdhA* strain (designation JW0713), and the parental strain K12 BW25113 were obtained from the E. coli Genetic Stock Center. The *ΔycgM ΔsdhA* double knockout strain was created by first excising the KanR selection marker from the *ΔycgM* strain. The KanR selection marker flanked by ~200 bp of sdhA flanking sequence was PCR amplified from genomic DNA isolated from the *ΔsdhA* strain using the 'Ec SdhA up200' and 'Ec SdhA down200' primers, and the amplicon was used to replace the native sdhA gene with the KanR marker in the *ΔycgM* strain (with excised KanR marker) using the lambda red recombination system. The knockout strains were PCR verified before use (Fig. S12). Strains were transformed with pUC19-ycgM, pUC19-sdhA, or empty vector by electroporation and transformants were selected on LB medium containing 100 μg mL$^{-1}$ ampicillin. For dilution spot assays, freshly prepared transformants were grown overnight in 5 mL of LB medium containing 100 μg mL$^{-1}$ ampicillin. Cells were washed with water, then serially diluted to $OD_{600} = 1.0, 0.2, 0.04, 0.008, 0.0016, 0.00032,$ and 0.000062. Five μL of each dilution was spotted on plates containing either LB medium, M9 minimal medium with 0.2% (w/v) glucose, M9 medium with 0.4% (w/v) glycerol, or M9 medium with 20 mM OAA. Plates were incubated at 22 °C for either 12 h (LB medium), 26 h (M9 glucose) or 48 h (M9 glycerol and M9 OAA) and high resolution images were acquired with a Biorad Gel Doc XR Imaging System. For growth curves, freshly prepared transformants were grown overnight in 5 mL of M9-glucose medium. Cells were washed with water, then used to inoculate 3 mL of fresh medium to $OD_{600} = 0.05$. Cultures were grown at 37 °C with shaking and $OD_{600}$ was recorded every 60 min. To determine colony size on plates, fresh transformants were grown overnight in LB medium containing 100 μg mL$^{-1}$ ampicillin and aliquots of serially diluted culture expected to contain ~100 cells (based on $8 \times 10^8$ cells mL$^{-1}$ $OD_{600}^{-1}$) were spread on plates containing LB medium. Plates were incubated at 37 °C for 16 h and high-resolution images were acquired with a Biorad Gel Doc XR Imaging System. ImageJ software was used to measure the size of 95 randomly selected colonies for each strain using the 'analyze particles' function.

*M. maripaludis* strain JJ with an in-frame deletion of the gene encoding uracil phosphoribosyltransferase was used as the wild-type strain in all experiments. Strains were grown in McCas medium or McCas-formate medium. Initial attempts to generate a mutant with an in-frame deletion of the gene encoding OAT1 were unsuccessful, presumably due to a growth defect associated with the mutant phenotype. The gene encoding OAT1 is adjacent to the gene encoding hypoxanthine phosphoribosyltransferase (*hpt*) on the chromosome; therefore, to generate the mutant, a deletion construct targeting both OAT1 and *hpt* was introduced followed by selection on medium containing 0.25 mg mL$^{-1}$ 8-azahypoxanthine. Under these conditions, a strain lacking OAT1 was successfully generated. The mutant strain had a growth defect (Fig. 5F), consistent with our inability to generate a mutant without the additional 8-azahypoxanthine-based selection. This phenotype could be easily suppressed upon serial transfer, so growth curves were always performed with strains as soon as possible after revival from storage at −80 °C.

To generate *M. maripaludis* lacking OAT1 and *hpt*, genomic regions flanking these genes were PCR amplified with primer pairs hpt9410-us-Fg and hpt9410-us-Rg to amplify a region upstream of the genes and hpt9410-ds-Fg and hpt9410-ds-Rg to amplify a region downstream of these genes. The resulting PCR products were placed into XbaI and NotI digested pCRuptneo by Gibson assembly. The plasmid was introduced into *M. maripaludis* via the polyethylene glycol (PEG) method of transformation. Transformants were selected on medium containing neomycin (1 mg mL$^{-1}$) and mutants selected on agar medium containing 8-azahypoxanthine. The mutant was complemented by introduction of OAT1 on the replicative vector pLW40neo, which was introduced into *M. maripaludis* by the PEG method of transformation.

## Metabolomics analysis

Freezer stocks of wild type (BW25113) and *ΔycgM E. coli* were streaked on LB plates and 4 single colonies from each strain were grown for 24 h in M9 minimal medium (0.2% glucose). Cells were washed with 1x M9 salts and used to inoculate 5 mL culture tubes containing M9 medium with either 0.2% glucose (natural isotopic abundance) or 0.2% $^{13}C_6$-glucose (fully labeled) to an initial $OD_{600} = 0.05$. Four replicate cultures of each genotype were grown in each media formulation by shaking at 37 °C for ~4 h until cells reached mid-log growth phase ($OD_{600}$ ~ 0.6; determined with a Thermo Scientific Genesys 30 spectrophotometer that can measure OD inside culture tubes). A rapid harvesting protocol was used to minimize metabolic disturbances[62]. Briefly, an equivalent of 1 mL at $OD_{600} = 1.0$ was quickly transferred to 1.5 mL polypropylene tubes, cells were pelleted in a microcentrifuge at full speed for 30 s, the supernatant was quickly aspirated and collection tubes were immediately snap frozen in liquid nitrogen. Samples were stored at −80 °C prior to extraction. After collecting the mid-log samples, cultures were shaken for an additional ~2 h until early stationary growth phase ($OD_{600}$ ~ 1.2) and samples were harvested as above. Collection tubes were placed on dry ice, 0.2 mL of cold 90% methanol was added, and tubes were incubated at −80 °C for 72 h. Samples were removed from the freezer, vortexed for 15 s, and incubated on ice for 3 h with vortexing every ~30 min. Afterwards samples were spun for 15 min in a microcentrifuge (16,000×*g*) at 4 °C and supernatants were collected for analysis.

Metabolomic data were obtained using an ultra-performance liquid chromatography-electrospray ionization-hybrid quadrupole-orbitrap mass spectrometer (Ultimate® 3000 HPLC, Q Exactive™, Thermo Scientific) with an autosampler and with a sample vial block maintained at 4 °C. Chromatographic separations were carried out on an SeQuant® ZIC®-cHILIC 3 μm, 100 Å, 100 ×2.1 mm column (Merck, Darmstadt, Germany) with column temperature 40 °C, flow rate 0.40 mL/min, and a 2 μL injection volume. Mobile phases A: 0.1%

formic acid in water and B: 0.1% formic acid in acetonitrile were delivered over a 23 min. gradient according to the following profile: initial 98% B, 2 min 98% B, 20 min 40% B, 22 min 98% B, 23 min 98% B. The MS conditions used were full scan (mass range 50-750 $m/z$, and 115-1000 $m/z$ in separate analyses), resolution 70,000, desolvation temperature 350 °C, spray voltage 3800 V, auxiliary gas flow rate 20, sheath gas flow rate 50, sweep gas flow rate 1, S-Lens RF level 50, and auxiliary gas heater temperature 300 °C. Xcalibur™ software version 2.1 (Thermo Scientific) was used for data collection.

Tandem MS data were obtained using data-dependent Top N acquisition (Full MS & dd-MS/MS). Precursor ions (top 5 most abundant ions per scan) were sequentially fragmented in the HCD collision cell with normalized collision energies (NCE) of 10, 20, 30, 40, 50, and 60 for six independent injections of each sample. MS/MS scans were acquired with 17,500 resolution, target value $1.0 \times 10^5$, 50 ms maximum injection time, and isolation window of 4.0 $m/z$. Data files were converted from.RAW to.mzML and.mgf formats using the ProteoWizard tool MSConvertGUI[63]. MZmine 2.53 was utilized for extracting exact-mass chromatographic data for isotope ratio calculations, for generating untargeted metabolite feature tables, and for tabulating peak heights from targeted exact masses for calculating relative signal intensity ratios[64].

Targets for quantification were selected two ways, first MZmine was used to generate separate untargeted feature tables for unmixed unlabeled and $^{13}$C-labeled samples. Those lists consisted of the $^{12}$C- or $^{13}$C-monoisotopic mass, retention time, and intensity values for each subset of samples. The lists were compared to find features within ±0.1 retention time on both lists with a $^{12}$C to $^{13}$C mass difference corresponding to an integer multiple of 1.003355 amu (the mass difference between $^{12}$C and $^{13}$C). These pairs of features were then sorted by difference in average intensity ratios between WT and Δ*ycgM* replicates at mid log and early stationary phase to identify metabolites that were changing with genotype. Elemental composition was calculated using both the $^{12}$C and the $^{13}$C monoisotopic masses considering only formulae that were present on both lists as a way to increase accuracy[65]. Metabolite identities (Supplementary Data file 1, tab1) were assigned to elemental compositions using prior identification in *E. coli* K12 (E. coli Metabolome DataBase, ECMDB)[66] and MS/MS spectral matches (described below). The second target selection strategy sought to include metabolites in the TCA cycle and central metabolism as well in pathways that included metabolites identified using the first approach. From this list of metabolites elemental compositions were obtained, $^{12}$C and $^{13}$C monoisotopic masses were calculated, extracted ion chromatograms (EICs) were compared to confirm co-elution, and isomeric metabolites were identified using the ECMDB. These lists were further constrained using MS/MS spectral assignments and were added to "Supplementary Data 1". Relative quantification between WT and Δ*ycgM* samples was accomplished using two complementary strategies. The first strategy is summarized in detail in "Supplementary Data 2" for pyruvate, starting with the 'Read Me' tab. Briefly, one first determines the ratio of $^{12}$C to $^{13}$C monoisotopic peaks in samples spiked with an equal volume of pooled samples with the other label for use in subsequent calculations. This is accomplished by plotting the EIC for the $^{12}$C-monoisotopic mass vs. the EIC for the $^{13}$C-monoisotopic mass for each sample-pool-mix, performing a linear regression, and using the slope of the regression as the $^{12}$C to $^{13}$C ratio. The regression strategy works well because the two isotopomers will coelute exactly even if the peak shape is not ideal. With the ratios in hand, one can derive the relative abundance of a specific compound by taking the ratio of two ratios [e.g. ($^{12}$C-WT@mid log/$^{13}$C-mid log pool) / ($^{12}$C-Δ*ycgM*@mid log/$^{13}$C-mid log pool) = WT/Δ*ycgM*@mid log]. This use of the internal isotope labeled metabolites provides an ideal internal control for ion suppression and other matrix effects that can introduce systematic error in complex metabolomics samples. A reciprocal labeling strategy is used throughout so that relative abundances are calculated using $^{12}$C-labeled samples mixed with $^{13}$C-pool in quadruplicate and then again using $^{13}$C-labeled samples mixed with $^{12}$C-pools. This labeling strategy makes it possible to eliminate the possibility of either very rare $^{13}$C-isotope effects or less rare perturbations derived from different contaminants in labeled vs. unlabeled reagents. The relative abundance by isotope ratio results and statistics are summarized in Supplementary Data 1, tab2. The second strategy derives the relative abundance by calculating the ratio of the average peak heights for each compound in all of the unmixed samples using the appropriate carbon isotope monoisotopic mass. These values were extracted using MZmine only if a peak was detectable with signal at least 5 times background. As $^{13}$C-enrichment in labeled samples was ~99%, the same as $^{12}$C-enrichment in natural abundance samples, no isotopic enrichment correction was needed. The relative abundance by peak height results and statistics are summarized in the third worksheet of Supplementary Data 1, worksheet 3, and while they are less accurate than the isotope ratio quantities they provide a reasonable abundance ratio approximation in cases where the isotope ratio regression or calculations failed either due to excessive noise or large abundance differences between treatments where the regression slope approaches 0 or infinity.

Molecular networking was used to cluster MS/MS from data collected at different collision energies using Global Natural Product Social Molecular Networking hosted at: gnps.ucsd.edu[67]. Raw data files were converted to the.*mgf* format, uploaded to the GNPS site, and filtered by removing all MS/MS fragment ions within ±17 Da of the precursor $m/z$. MS/MS spectra were window filtered by choosing only the top 6 fragment ions in the ±50 Da window throughout the spectrum. The clustered spectra were then searched against GNPS spectral libraries and all of the matches kept between network spectra and library spectra were required to have a score >0.5 and at least 2 matched peaks. This atypical setting for the number of matching peaks was selected to avoid exclusion of small and phosphorylated metabolites that often have simple MS/MS spectra (Supplementary Data 1, worksheets 5 and 6).

## Statistics and reproducibility

No statistical method was used to predetermine sample sizes. No data were excluded from the analyses. Metabolomics samples were randomly analyzed with regularly interspaced blanks and pooled samples to control for analytical drift; other experiments were not randomized. Investigators were blinded while preparing and analyzing metabolomics samples; investigators were not blinded to allocation during other experiments and outcome assessment. Independently and freshly prepared enzyme preparations were used in all enzyme assay replicates. To the extent possible, spectrophotometric assays were conducted in parallel and master mixes were used to limit any potential bias and variation.

## Reporting summary

Further information on research design is available in the Nature Portfolio Reporting Summary linked to this article.

## Data availability

The metabolomics data generated in this study have been deposited to both the Metabolights database under accession code MTBLS8848 and the Data Repository for U of M under accession code 259320. The biochemical and physiological data generated in this study are provided in the Supplementary Information/Source Data file. Source data are provided with this paper.

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

## Acknowledgements

We thank Dallas Dornink, Lead Processing Technician at the University of Minnesota, for providing freshly harvested beef hearts. Support was provided by startup funds from the University of Minnesota, Department of Plant and Microbial Biology (Niehaus) and from the U.S. Department of Energy, Office of Science, Basic Energy Sciences under grant number DE-SC0019148 (Costa).

## Author contributions

T.D.N.: conceptualization, methodology, investigation, formal analysis, supervision, and writing. K.C.C.: investigation and formal analysis. X.K.: investigation and formal analysis. A.J.Z.: investigation and formal analysis. K.B.W.: investigation and formal analysis. K.F.S.: investigation and formal analysis. A.D.H.: methodology, investigation, and formal analysis.

## Competing interests

The authors declare no competing interests.
