## [Peer Review File · Nature Communications]

A universal metabolite repair enzyme removes a strong inhibitor of the TCA cycleEditorial Note: This manuscript has been previously reviewed at another journal that is not operating a transparent peer review scheme. This document only contains reviewer comments and rebuttal letters for versions considered at *Nature Communications*.

REVIEWERS' COMMENTS

Reviewer #1 (Remarks to the Author):

This paper is much improved over the previous paper. Specifically, the authors addressed all of my concerns (very carefully) and added significant new results (specifically, the genetics approach using knockout cells and metabolomics analysis). I would support publication.

A few suggestions – I would tone down the conclusions

Last line of Abstract – replace “indicate” with “suggest” or “are consistent with”.

p. 7 – is it really time for renaming? (again, maybe suggest)

Maybe add as scheme/figure with the reaction – malate/oxidation to OAA

Reviewer #2 (Remarks to the Author):

Based on the reviewers' comments, the authors have included important new experimental data in the revised manuscript, which strengthen it considerably and provide better support to the in vivo existence and relevance of the newly proposed metabolite damage/repair pathway.

In particular, I can state that all of my previous comments were addressed in a satisfactory manner and I have only a few remaining minor comments (listed below). Unless the other reviewers request it, I would not need to see another revised version of this manuscript.

Remaining minor comments:

- The authors mention twice (L46 and L272-3) that the malate oxidation side activity of SDH is the most promiscuous activity associated with the TCA cycle. I don't fully get the meaning of that affirmation and if the authors refer to the ratio of main activity to side activity, I'm not sure how one can claim this as a fact, when there are maybe side activities for other enzymes that have not been identified or quantified yet. I would recommend to reformulate this statement for better clarity and for leaving some space to 'until we find another enzyme with a potentially even higher side activity'.

- L110-2: it would be good to briefly explain based on what it is known that residues D102 and R106 are active site residues in FAHD1. In addition, it would be good to state that the M2-FAHD1 enzyme was assayed and add the results to the table in Fig. 2D (showing low activity values if still detectable or 'ND' if not detectable).

- I applaud the authors for the carefully conducted additional metabolomics analyses, which strengthen the paper. I was, however, surprised to not see an accumulation of succinate in the *ycgM* KO strain (on the contrary, its levels are significantly decreased compared to WT). The authors should mention it and propose plausible explanations. In the description of the metabolomics results, the authors also mention another study measuring OAA levels in another *E. coli* mutant. If I understand correctly, OAA was not detected in the metabolomics analyses presented here? Making an extra effort to measure OAA levels could have further strengthened the hypotheses put forward in this manuscript.

Our responses (colored blue) to reviewer comments are listed below:

Reviewer #1 (Remarks to the Author):

This paper is much improved over the previous paper. Specifically, the authors addressed all of my concerns (very carefully) and added significant new results (specifically, the genetics approach using knockout cells and metabolomics analysis). I would support publication.

A few suggestions – I would tone down the conclusions

We toned down the conclusions in both the abstract and discussion, including the two instances below.

Last line of Abstract – replace “indicate” with “suggest” or “are consistent with”.

We replaced indicate with suggest.

p. 7 – is it really time for renaming? (again, maybe suggest)

We toned down the conclusions by removing two sentences proposing a rename – we agree that this may be unnecessary.

Maybe add as scheme/figure with the reaction – malate/oxidation to OAA

We added a new figure (Fig 1) showing a scheme of the metabolite damage repair pathway.

Reviewer #2 (Remarks to the Author):

Based on the reviewers' comments, the authors have included important new experimental data in the revised manuscript, which strengthen it considerably and provide better support to the in vivo existence and relevance of the newly proposed metabolite damage/repair pathway. In particular, I can state that all of my previous comments were addressed in a satisfactory manner and I have only a few remaining minor comments (listed below). Unless the other reviewers request it, I would not need to see another revised version of this manuscript.

Remaining minor comments:

- The authors mention twice (L46 and L272-3) that the malate oxidation side activity of SDH is the most promiscuous activity associated with the TCA cycle. I don't fully get the meaning of that affirmation and if the authors refer to the ratio of main activity to side activity, I'm not sure how one can claim this as a fact, when there are maybe side activities for other enzymes that have not been identified or quantified yet. I would recommend to reformulate this statement for better clarity and for leaving some space to 'until we find another enzyme with a potentially even higher side activity'.

We reformulated the statements on L46 and L272-3 to replace “most promiscuous” with “a major promiscuous activity” and qualify that this is the most promiscuous reaction *known to*

occur in the TCA cycle.

- L110-2: it would be good to briefly explain based on what it is known that residues D102 and R106 are active site residues in FAHD1. In addition, it would be good to state that the M2-FAHD1 enzyme was assayed and add the results to the table in Fig. 2D (showing low activity values if still detectable or 'ND' if not detectable).

We now explain that residues D102 and R106 are predicted to mediate substrate binding and provide two references. We also added a sentence stating that M2-FAHD1 had <0.1% of the activity as non-mutated human FAHD1 and show the data in supplemental figure 4.

- I applaud the authors for the carefully conducted additional metabolomics analyses, which strengthen the paper. I was, however, surprised to not see an accumulation of succinate in the ycgM KO strain (on the contrary, its levels are significantly decreased compared to WT). The authors should mention it and propose plausible explanations. In the description of the metabolomics results, the authors also mention another study measuring OAA levels in another E. coli mutant. If I understand correctly, OAA was not detected in the metabolomics analyses presented here? Making an extra effort to measure OAA levels could have further strengthened the hypotheses put forward in this manuscript.

We were also expecting succinate levels to increase in the KO strain and were initially surprised to see that succinate significantly decreased in the KO. However, as we expanded our analysis we saw evidence for increased use of anaerobic pathways and evidence that the TCA cycle is functioning anaplerotically (essentially operating in the reverse or reductive direction). This could explain the result; as lower succinate would be expected if SDH activity was inhibited during anaplerosis. We added a sentence to the metabolomics paragraph to discuss this explanation.

Flux analysis is needed to confirm this interpretation and we are planning to measure flux in the future (as well as measure OAA levels reliably). While this data would support the conclusions of this study, we believe it is a bit beyond the scope and is better suited as a future study.